# Vitamin D Receptor Protects against Radiation-Induced Intestinal Injury in Mice via Inhibition of Intestinal Crypt Stem/Progenitor Cell Apoptosis

**DOI:** 10.3390/nu13092910

**Published:** 2021-08-24

**Authors:** Wusun Li, Yingying Lin, Yujia Luo, Yuqi Wang, Yao Lu, Yixuan Li, Huiyuan Guo

**Affiliations:** College of Food Science and Nutritional Engineering, China Agricultural University, 17 Tsinghua East Road, Beijing 100083, China; 18137720201@163.com (W.L.); linyingying219@163.com (Y.L.); luoyujia000@126.com (Y.L.); wangyuqi0715@163.com (Y.W.); lylywhm@126.com (Y.L.); liyixuan9735@126.com (Y.L.)

**Keywords:** apoptosis, crypt stem/progenitor cell, ionizing radiation, Pmaip1, vitamin D/vitamin D receptor

## Abstract

It is urgent to seek new potential targets for the prevention or relief of gastrointestinal syndrome in clinical radiation therapy for cancers. Vitamin D, mediated through the vitamin D receptor (VDR), has been identified as a protective nutrient against ionizing radiation (IR)-induced damage. This study investigated whether VDR could inhibit IR-induced intestinal injury and explored underlying mechanism. We first found that vitamin D induced VDR expression and inhibited IR-induced DNA damage and apoptosis in vitro. VDR was highly expressed in intestinal crypts and was critical for crypt stem/progenitor cell proliferation under physiological conditions. Next, VDR-deficient mice exposed to IR significantly increased DNA damage and crypt stem/progenitor cell apoptosis, leading to impaired intestinal regeneration as well as shorter survival time. Furthermore, VDR deficiency activated the Pmaip1-mediated apoptotic pathway of intestinal crypt stem/progenitor cells in IR-treated mice, whereas inhibition of Pmaip1 expression by siRNA transfection protected against IR-induced cell apoptosis. Therefore, VDR protects against IR-induced intestinal injury through inhibition of crypt stem/progenitor cell apoptosis via the Pmaip1-mediated pathway. Our results reveal the importance of VDR level in clinical radiation therapy, and targeting VDR may be a useful strategy for treatment of gastrointestinal syndrome.

## 1. Introduction

Gastrointestinal (GI) syndrome, with 60–80% incidence occurrence in radiation therapy for abdominal or pelvic cancers, is a lethal disorder and an obstacle to cancer cure [1]. Therefore, it is especially crucial to seek new potential targets for the prevention or mitigation of ionizing radiation (IR)-induced intestinal injury. Epidemiological and experimental studies have indicated that vitamin D (VD) acts as a protective nutrient against IR-induced skin and bone marrow damage and also prevents harmful effects for cancer patients receiving radiation therapy [2,3,4]. It is suggested that VD deficiency increases susceptibility to gastrointestinal injury [5]. A prospective observational study reported that VD deficiency could aggravate acute proctitis in cancer patients with radiotherapy [6]. However, the study on VD protection against IR-induced intestinal injury is very limited.

The underlying mechanism of VD on IR-induced intestinal injury is unclear. It may be related to regulating gut microbiota and inhibiting cell death by reducing ROS production via vitamin D receptor (VDR) [4,7]. VDR, a nuclear receptor, mediates the biological functions of vitamin D [8]. VDR exists in a variety of cells and is highly expressed in the intestine. VD acts as a ligand to bind and activate VDR to stimulate gene expression related to intestinal calcium and phosphate absorption, skeletal and calcium homeostasis, hair cycle regulation, immune modulation and cancer prevention [9]. VD/VDR plays a major role in intestinal cell differentiation, barrier function, immunity as well as host–microbe interactions [10]. Clinical trials have shown that VD deficiency and low VDR level are highly prevalent in patients with inflammatory bowel diseases who are vulnerable to GI syndrome, suggesting that GI syndrome may be associated with low VD/VDR level [11,12]. Evidence from animal studies suggested that VDR knockout mice aggravated the symptoms of colitis [13,14,15]. In addition, overexpression of VDR in colorectal epithelium could alleviate colitis [16]. These findings suggest that VDR exerts protective effect against inflammation-induced colonic injury and may inhibit IR-induced intestinal injury.

Intestinal crypt stem/progenitor cells display a remarkable capacity for intestinal renewal and repair [17]. Some researchers believe that crypt stem/progenitor cell apoptosis is the primary factor initiating IR-induced intestinal injury [18,19,20]. Therefore, inhibition of intestinal stem/progenitor cell apoptosis has become a new target for reducing IR-induced intestinal injury. We hypothesize that VDR can inhibit intestinal stem/progenitor cell apoptosis based on the following evidence: First, VD/VDR is necessary for intestinal stem cell functions of renewal activity [21,22]. Second, intestinal epithelial VDR deficiency impairs the function of intestinal crypt Paneth cells, which are important for supporting intestinal stem cells [14,23,24]. Third, VD/VDR can inhibit colorectal epithelial cell apoptosis to protect inflammation-induced colonic injury [15,25]. We hypothesize that VDR can attenuate intestinal stem/progenitor cell apoptosis to further protect against IR-induced intestinal injury.

In our research, we employed VDR knockout (KO) mice to assess the effect of VDR on apoptosis of intestinal crypt stem/progenitor cells under physiological conditions and IR exposure. Then, we investigated the function of VDR on IR-induced intestinal injury. In addition, the mechanism of VDR inhibiting IR-induced intestinal crypt stem/progenitor cell apoptosis was explored.

## 2. Materials and Methods

### 2.1. Cell Culture and VD Treatment

Rat small intestinal crypt epithelial cell line (IEC-6) was purchased from iCell Bioscience Inc. (Shanghai, China). The cells were maintained in Dulbecco’s modified Eagle’s medium (Thermo Fisher, Waltham, MA, USA) with 10% (*v*/*v*) heat-inactivated FBS (Biological Industries, Beit Hamek, Israel) and 1% penicillin–streptomycin (Beyotime, Shanghai, China) at 37 °C in a 5% CO_2_ incubator. The IEC-6 cells were treated with 10 nM 1,25(OH)_2_D_3_ (Sigma-Aldrich, St. Louis, MO, USA), the bioactive form of VD. The cells were then exposed to 12 Gy ^60^Co γ-radiation (1 Gy/min). After 24 h, the protein samples were collected.

### 2.2. Cell Apoptosis Assay

IEC-6 cells were inoculated in triplicate in 6-well plates, incubated for 24 h then treated with 0 or 10 nM 1,25(OH)_2_D_3_ for 6 h prior to 12 Gy IR. After 24 h, cell apoptosis was detected using an Annexin V-FITC Apoptosis Detection Kit (Beyotime, Shanghai, China). The measurements were performed based on the manufacturers’ protocols. The pictures were obtained with a fluorescence microscope (Carl Zeiss, Jena, Germany). At least 10 fields of view were captured, and 1500 total cells were assessed in each group. The quantification of fluorescent cells was detected using Image J software.

### 2.3. Cell Immunofluorescence

For cell immunofluorescence staining, the cells were fixed with 4% paraformaldehyde in PBS for 15 min and incubated with 0.1% Triton X 100 for 20 min. Then the cells were blocked with 5% BSA for 1 h. After incubation with anti-γH2AX antibody (1:400, 9718T, Cell Signaling Technology, Danvers, MA, USA) overnight at 4 °C, the cells were incubated with secondary antibodies for 1 h, and then the cells were stained with DAPI (C1005, Beyotime, Shanghai, China). The images were captured with a fluorescence microscope (Carl Zeiss, Jena, Germany). At least 10 fields of view were captured, and 1500 total cells were assessed in each group. The quantification of fluorescent cells was detected by the Image J software.

### 2.4. Animals and IR

C57/BL6 VDR heterozygote mice were purchased from Beijing Biocytogen Co., Ltd. (Beijing, China). VDR wild-type (WT) and VDRKO mice were produced through VDR heterozygote breeding. Both VDRKO and WT littermates were fed from weaning on a high-calcium, high-lactose rescue diet (2% Ca, 1.25% P, 20% lactose and 1300 IU/kg vitamin D_3_) for their survival [26]. The specific composition of the rescue diet was shown in Appendix A. Diet and water were provided ad libitum. All mice were exposed to a 12-h light–dark cycle in an SPF barrier facility controlled for temperature (23 °C ± 2 °C) and relative humidity (40–70%). Mice were whole-body irradiated at dose of 12 Gy IR with a ^60^Co source (Peking University, Beijing, China). All animal experimental procedures were evaluated and authorized by the Regulations of Beijing Laboratory Animal Management and approved by the Institutional Animal Care and Use Committee of China Agricultural University (Permit number: Aw51018102-4-3).

### 2.5. Histology, Immunohistochemistry and Immunofluorescence

Intestinal tissues were freshly isolated and fixed with 4% paraformaldehyde before paraffin embedding. After deparaffination and rehydration, the sections were stained with hematoxylin and eosin (H&E). For immunohistochemistry, tissues were treated with 0.01 M citrate buffer (pH 6.0) in a microwave and then blocked with blocking buffer (Beyotime, Shanghai, China) for 1 h. After incubation with primary antibodies at 4 °C overnight, sections were then immunostained using the SP method. For immunofluorescence staining, after incubation with the primary antibodies, the sections were incubated with Cy3-labeled Goat Anti-Rabbit IgG (1:1000, A0516, Beyotime, Shanghai, China) and counterstained with DAPI. The slides were cover slipped, and the stainings were observed by using a fluorescence microscope (Leica DM4/6B, Leica, Germany). The following primary antibodies were used: anti-VDR (1:200, 12550S, Cell Signaling Technology, Danvers, MA, USA), anti-Ki67 (1:200, ab16667, Abcam, Cambridge, MA, USA), anti-Sox9 (1:100, ab185230, Abcam, Cambridge, MA, USA), anti-Cleaved Caspase-3 (1:2000, 9664S, Cell Signaling Technology, Danvers, MA, USA) and anti-γH2AX (1:400, 9718T, Cell Signaling Technology, Danvers, MA, USA). At least 10 fields of view were captured, and 100 intact crypts were assessed per mouse. The crypt depth and immunopositive cells were quantified using Image J software.

### 2.6. RNA Isolation and Real-Time Quantitative PCR

Total RNA was isolated from intestinal tissues using TRIzol reagent (Invitrogen, Carlsbad, CA, USA) according to the manufacturer’s protocols. cDNA was synthesized using a 5× All-In-One MasterMix Kit (Applied Biological Materials, Vancouver, BC, Canada), and real-time quantitative PCR (RT-qPCR) was then performed using SYBR Premix Ex Taq (Takara, Tokyo, Japan). The relative expression levels were normalized to those of *Gapdh.* The sequences of primers are shown in Table 1.

### 2.7. Western Blotting (WB)

Total protein was extracted from intestinal tissues or IEC-6 cells using RIPA lysis buffer (Beyotime, Shanghai, China) and 1% PMSF (Beyotime, Shanghai, China). Denatured protein samples were separated with 10% SDS-PAGE and transferred to 0.45 μm polyvinylidene difluoride membranes (Millipore, Billerica, MA, USA). The membranes were incubated with primary antibodies overnight at 4 °C. After washing, the membranes were incubated with a 1:5000 dilution of HRP conjugated secondary antibodies. The following primary antibodies were used: anti-VDR (1:2000, 12550S, Cell Signaling Technology, Danvers, MA, USA), anti-Pmaip1 (1:500, ab13654, Abcam, Cambridge, MA, USA), anti-Cleaved Caspase-3 (1:2000, 9664S, Cell Signaling Technology, Danvers, MA, USA) and anti-β-Actin (1:1000, 4970S, Cell Signaling Technology, Danvers, MA, USA). The Western blot bands were quantified using Image J software, and the quantitative results were expressed by integral optical density (IOD).

### 2.8. TUNEL Assay

TUNEL staining was detected using an One Step TUNEL Apoptosis Assay Kit (Beyotime, Shanghai, China), according to the manufacturer’s protocols. Briefly, after deparaffination and rehydration, paraffin sections were incubated with the TUNEL reaction mixture at 37 °C for 1 h. Then the sections were stained with DAPI and the images were obtained by using a fluorescence microscope (Carl Zeiss, Jena, Germany). TUNEL-positive cells were counted from a minimum of 10 fields and 80 intact crypts per mouse using Image J software.

### 2.9. Isolation of Crypt Stem/Progenitor Cells

The intestinal crypts were isolated according to previous methods [27]. Briefly, the intestines were opened longitudinally and cut into 2 mm fragments. The tissue pieces were cultured in Gentle cell dissociation reagent (STEMCELL Technologies, Vancouver, BC, Canada) at 40 rpm for 15 min at 4 °C. Then the supernatant was removed, and the sediment was washed with PBS. After mechanical disruption by pipetting, the supernatant was passed through a 70 μm strainer and centrifuged at 1000 rpm for 5 min to collect crypts.

### 2.10. siRNA Transfection

The siRNA targeting *Pmaip1* (si-Pmaip1) was transfected into IEC-6 cells by using lipofectamine 2000 (lipo) kit (Invitrogen, Carlsbad, CA, USA) medium. Briefly, the lipo solution contained 5 μL lipo and 100 μL Opti-MEM (Invitrogen, Carlsbad, CA, USA) medium. The siRNA solution contained 5 μL si-Pmaip1 and 100 μL Opti-MEM. The mixture, containing the lipo solution and siRNA solution, was added to the cells at 70% confluence, and 6 h after transfection, the medium was subsequently replaced with a complete medium. The transfection efficiency was observed immediately by a fluorescence microscope. The cells were transfected for 12 h, followed by 12 Gy IR. After 24 h, protein samples were collected, and cell apoptosis was detected using an Annexin V-FITC Apoptosis Detection Kit. The *Rattus norvegicus*-si-Pmaip1 sequences are listed in Appendix A.

### 2.11. Statistical Analysis

All the experiments were performed at least in triplicate. Statistical analyses were performed with SPSS version 23.0 (IBM, Armonk, NY, USA). Data was statistically analyzed using Student’s t test. In cases where data were not normally distributed, Mann–Whitney U tests were used. When multiple factors were included, the main effects and interaction of two factors (IR × VD, genotype × time) were analyzed by univariate two-factor ANOVA. Mean comparisons were analyzed using Tukey’s post-hoc test when the main effect was significant. When an interaction was found, mean comparisons were conducted conditionally. The results were presented as mean ± standard deviation (SD). Differences at *p* < 0.05 were considered as significant.

## 3. Results

### 3.1. VD/VDR Attenuates IR-Induced DNA Damage and Apoptosis in IEC-6 Cells

γH2AX is a biomarker of cellular radiosensitivity and DNA damage [28]. Non-IR cells showed a weak positive signal, whereas the γH2AX-positive cells were significantly increased at 1 h after 12 Gy IR (Figure 1A). VD treatment significantly reduced the γH2AX-positive cells after IR (Figure 1A,B). The proportion of Annexin V-positive cells was very small in non-IR IEC-6 cells, and VD treatment did not affect cell apoptosis (Figure 1C,D).

Following IR, the apoptotic cells were markedly increased compared with non-IR cells, while VD treatment effectively reduced IR-induced apoptosis (Figure 1C,D). We examined the VDR expression and found that VD effectively increased the protein level of VDR following IR (Figure 1E). Meanwhile, apoptotic protein Cleaved Caspase-3 (Casp3) expression was significantly decreased with VD treatment (Figure 1E). These results indicated that VD/VDR was crucial for amelioration of IR-induced DNA damage and apoptosis in IEC-6 cells.

### 3.2. VDR Is Highly Expressed in Intestinal Crypts and Significantly Upregulated in Response to IR

We investigated the expression pattern of VDR in mice intestine by immunohistochemistry (IHC) and immunofluorescence (IF). VDR was highly expressed in intestinal crypts (Figure 2A,B). Next, the changes of VDR levels exposed to IR were examined by RT-qPCR and WB. In an apoptosis phase of 5 h after IR, VDR expression was significantly upregulated at mRNA and protein levels (Figure 2C,D). However, in an intestinal regeneration phase of 3–4 d after IR, we detected a drop in VDR expression (Figure 2C,D). In a homeostasis phase of 5 d after IR, VDR expression did not recover to the physiological level (Figure 2C,D). These results indicated that VDR was responsive to IR and VDR activation might play an important role in the apoptosis phase after IR.

### 3.3. VDR Deficiency Impairs Intestinal Structure and Crypt Stem/Progenitor Cell Proliferation

Hypothesizing that VDR upregulation may inhibit IR-induced intestinal damage, we generated a loss of function model of VDRKO mouse by breeding of VDR heterozygotes. First, we studied the function of VDR in the intestine under physiological condition. The body weights and intestinal lengths of VDRKO mice were smaller than WT mice (Figure 3A,B). The VDR expression could not be detected in intestines of VDRKO mice by WB (Figure 3C). H&E staining results showed that crypt depth in VDRKO mice was significantly reduced (Figure 3D). Ki67 is a marker of proliferating cells, including intestinal stem cells and progenitor cells [29]. VDR deficiency significantly reduced the Ki67^+^ cells per crypt (Figure 3E). SOX9 is a marker of intestinal stem/progenitor cells [30]. The SOX9^+^ cells were markedly decreased in VDRKO mice (Figure 3F). However, VDR knockout had no significant effect on apoptosis as determined by TUNEL staining (Figure 3G). These results indicated that VDR loss impaired intestinal structure and crypt stem/progenitor cell proliferation rather than induced apoptosis under physiological conditions.

### 3.4. VDR Deficiency Suppressed Intestinal Epithelial Regeneration Following IR

To evaluate the protective role of VDR against IR-induced intestinal injury, we examined the intestinal structural changes. Histological analysis revealed that WT and VDRKO mice displayed comparable intestinal architectures 5 h after IR (Figure 4A). However, there were fewer regenerated crypts in VDRKO mice at 3 d after IR exposure (Figure 4A). After 4 d following IR, the differences had narrowed. However, Ki67-positive cells in each regenerated crypt of VDRKO mice were significantly reduced at 3–5 d after IR (Figure 4B). The survival time of VDRKO mice was significantly shorter compared with WT mice (Figure 4C). These results showed that VDR was important for intestinal epithelial regeneration after IR.

### 3.5. VDR Deficiency Aggravates IR-Induced Crypt Stem/Progenitor Cell Apoptosis

To understand the intestinal injury resulting from VDR deficiency following IR, we investigated apoptotic cells in VDRKO mice. First, we investigated whether VDR deficiency affected DNA damage. The γH2AX^+^ cells were markedly increased in the crypt cells of VDRKO mice at 5 h post-IR (Figure 5A). In addition, apoptosis was increased in the entire intestine with higher Cleaved Casp3 expression in VDRKO mice (Figure 5B,C). Then, intestinal crypt cell apoptosis was detected, which is closely related to IR-induced intestinal injury [20]. The Cleaved Casp3^+^ cells in crypts were increased in VDRKO mice compared with WT mice (Figure 5D). TUNEL staining results showed that VDR deficiency significantly increased apoptosis in the crypts at 5 h post-IR (Figure 5E). Quantification of apoptotic cell position analysis revealed that occurrence of apoptosis was more frequent at the cell positions 2–4. However, the number of TUNEL^+^ cells was increased by more than 50% in VDRKO mice at the cell positions 3–6 compared with WT mice (Figure 5E). These results demonstrated that VDR was crucial to resist IR-induced crypt stem/progenitor cell apoptosis.

### 3.6. VDR Suppresses IR-Induced Crypt Stem/Progenitor Cell Apoptosis via Pmaip1-Mediated Pathway

To investigate the potential mechanisms of VDR inhibiting IR-induced intestinal crypt stem/progenitor cell apoptosis, we examined the expression of proteins associated with apoptosis in intestinal crypts of mice after IR. Pmaip1 is a key pro-apoptotic protein that has been shown to be activated by IR [31], so we detected whether VDR knockout has a role in IR-induced Pmaip1 activation in crypt stem/progenitor cells. The protein level of VDR was not detected in VDRKO mice (Figure 6A,B). We found the protein expression of Pmaip1, which is critical for IR-induced apoptosis [32], was significantly induced in crypt stem/progenitor cells of VDRKO mice at 5 h post-IR, as well as the Cleaved Casp3 expression (Figure 6A,C,D). These results indicated that Pmaip1-mediated pathway was activated at 5 h after IR in crypt stem/progenitor cells of VDRKO mice.

To further determine whether VDR was involved in Pmaip1-mediated pathway, the Pmaip1 expression was downregulated by si-Pmaip1 in IEC-6 cells, which exhibited good features of crypt stem/progenitor cells [33]. The transfection efficiency of si-Pmaip1 in IEC-6 cells was determined by fluorescence microscope (Appendix A). Among three sequences of si-Pmaip1, si-Pmaip1-3 exhibited high knockdown efficiency with 92% at mRNA expression level (Appendix A). The knockdown efficiency of si-Pmaip1-3 at protein level is presented in Figure 6F with 67%. Annexin V-positive cells was significantly reduced in the si-Pmaip1 group compared with control cells after IR (Figure 6E). Moreover, Cleaved Casp3 expression was suppressed by si-Pmaip1 following IR (Figure 6F). These results indicated that Pmaip1 was a crucial apoptotic mediator and might support the claim that VDR inhibited IR-induced intestinal crypt stem/progenitor cell apoptosis through the Pmaip1-mediated pathway.

## 4. Discussion

IR-induced intestinal injury is a leading limitation of radiation therapy for abdominal and pelvic tumors [1]. Intestinal crypt stem/progenitor cell apoptosis is a major cause and taken as a measure of intestinal injury [34]. Consequently, relief of crypt stem/progenitor cell apoptosis is vital to combat intestinal damage caused by radiation. The present study indicated that VDR could protect against IR-intestinal injury and prolong the survival time of mice via inhibition of crypt stem/progenitor cell apoptosis through modulation of Pmaip1 activity. To our knowledge, this is the first report that VDR can inhibit IR-induced crypt stem/progenitor cell apoptosis to protect against intestinal injury.

Intestinal crypt cell proliferation is critical to intestinal homeostatic responses and recovery from injury [35]. On the contrary, stem/progenitor cell proliferation defects can make the intestine sensitive to IR-induced damage [36,37]. In this study, the high expression of VDR in crypt cells enabled stem/progenitor cells to proliferate and maintain homeostasis. In addition, VDR deficiency led to smaller crypts and fewer proliferating stem/progenitor cells. This might explain why VDRKO mice were more vulnerable to IR-induced intestinal injury. Our results were in agreement with Peregrina et al., who reported that low dietary VD or inactivated VDR compromised the function of intestinal stem cells with fewer stem cell progeny [21]. A recent study using intestinal stem cell-derived organoids showed that VDR-deficient organoids lowered crypt cell proliferation [38]. Our data provided evidence that VDR was critical for modulation of stem/progenitor cell proliferation under physiological conditions.

Following exposure to high-dose IR, apoptosis is dramatically elevated in crypt stem/progenitor cells within a period of 3–6 h post-IR [18]. Regenerating crypts, originating from radio-resistant crypt base columnar cells that survive IR, rapidly form between 3 and 4 d after IR [18,39]. Within about 5 d, the intestine is fully restored [40]. The result that VDR was significantly upregulated in the apoptosis phase indicated that VDR might participate in the regulation of cell apoptosis. During the apoptosis phase, VDR acted as an anti-apoptotic factor to protect crypt stem/progenitor cells against IR-induced apoptosis. This was consistent with our finding that VD/VDR exhibited anti-apoptotic activity following IR in IEC-6 cells. Our results revealed the importance of the VDR level to protect stem/progenitor cells from apoptosis in clinical radiation therapy.

During the regeneration phase, surviving stem cells, with amazing regenerative capacity, proliferate to repopulate the crypt compartment [18,40]. Although it has been observed that VDR could enhance stem/progenitor cell proliferation under physiological conditions, the inhibitory effect of intestinal epithelial regeneration in VDRKO mice after IR was not caused by the inhibition of stem/progenitor cell proliferation because of the low VDR expression in the regeneration phase. It was possible that VDR deficiency caused more crypt stem/progenitor cell apoptosis in a single crypt following IR; as a result, fewer surviving stem cells proliferate to form regenerative areas and repair the damaged intestinal epithelia. In the homeostasis phase, the expression of VDR was elevated with the recovery of intestinal epithelia. However, the expression of VDR did not recover to the physiological level. This result might be explained by the inflammation occurrence caused by secondary effects in response to the initial radiation injury [1]. VDR expression is always low in intestinal inflammation, which is a chronic injury that is difficult to recover following exposure to IR [1,6]. Our findings support that high VDR expression is important after radiotherapy to protect IR-induced subsequent chronic intestinal injury.

Pmaip1 is a pro-apoptotic Bcl-2 homology domain 3-only protein [31]. It has been demonstrated that the Pmaip1-mediated pathway plays an important part in IR-induced apoptotic response [41,42,43]. Pmaip1 knockout mice showed resistance to IR-induced intestinal crypt cell apoptosis and survived longer than WT mice [32]. Our results indicated that the Pmaip1 pathway was activated in intestinal stem/progenitor cells of VDRKO mice at 5 h after IR, which may explain the higher occurrence of stem/progenitor cell apoptosis and shorter survival times of VDRKO mice. Likewise, the inhibition of Pmaip1 pathway activation in vitro alleviated IR-induced IEC-6 cell apoptosis. Kim et al. reported that suppression of Pmaip1 expression protected human lung epithelial cells from IR-induced apoptosis [42]. Our findings indicated that VDR protected against IR-induced intestinal stem/progenitor cell apoptosis by inhibiting the Pmaip1-mediated pathway. Further work is needed to investigate the functional study of Pmaip1 in vivo and whether VDR can directly regulate Pmaip1. In order to further prove the effect of VDR upregulation on IR-induced intestinal stem/progenitor cell apoptosis, establishing a mouse model with overexpression of VDR, such as generation of transgenic mice with overexpression of VDR in intestinal epithelium or dietary intervention of vitamin D, is worth further investigation in the future.

## 5. Conclusions

In summary, we demonstrated that VDR could regulate the Pmaip1-mediated pathway and diminish stem/progenitor cell apoptosis and further protect against intestinal injury induced by IR. Therefore, VDR might be a potential target for prevention and management of IR-induced intestinal injury.

## Figures and Tables

**Figure 1 nutrients-13-02910-f001:**
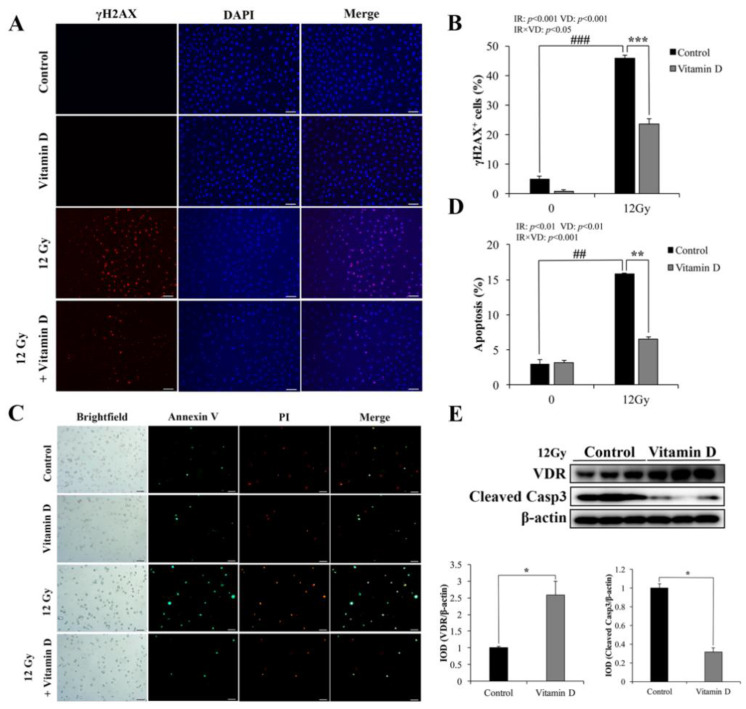
Vitamin D/Vitamin D receptor alleviates IR-induced DNA damage and apoptosis in IEC-6 cells. (**A**) Representative immunofluorescence (IF) images of γH2AX staining in IEC-6 cells at 1 h post 12 Gy IR pretreated 10 nM 1,25(OH)_2_D_3_. (**B**) Quantification of percentage of γH2AX^+^ cells was analyzed. (**C**) Representative IF images of Annexin V/PI staining in IEC-6 cells at 24 h post 12 Gy IR pretreated 10 nM 1,25(OH)_2_D_3_. (**D**) Quantification of apoptotic cells with Annexin V-positive signaling was analyzed. # Compared with the 0 Gy-control group; * compared with the 12Gy-VD group. (**E**) Vitamin D receptor (VDR) and Cleaved Caspase 3 (Casp3) in IEC-6 cells were determined by WB at 24 h after 12 Gy IR pretreated with 10 nM 1,25(OH)_2_D_3_ for 6 h. Quantitative analysis of Western blotting of VDR and Cleaved Casp3 expression. Values are means ± SD. * *p* < 0.05, ** *p* < 0.01, *** *p* < 0.001; ^##^ *p* < 0.01, ^###^ *p* < 0.001. Scale bar, 20 μm.

**Figure 2 nutrients-13-02910-f002:**
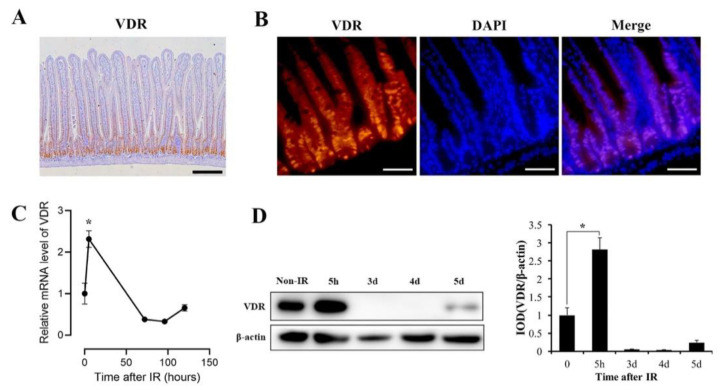
Intestinal VDR expression in mice under physiological condition and IR exposure. (**A**) Representative immunohistochemistry (IHC) image of VDR staining in intestines of mice. Scale bar, 200 μm. (**B**) Representative IF pictures of VDR expression of mouse intestine. Scale bar, 50 μm. (**C**) VDR mRNA expression in the intestines of mice at 0, 5 h, 72 h, 96 h and 120 h after 12 Gy IR was evaluated by RT-qPCR. *n* = 6 per group. (**D**) VDR protein expression in intestines of mice at different times of 0, 5 h, 3 d (days), 4 d and 5 d after 12 Gy IR was assessed by WB. Quantitative analysis of WB of VDR expression. Values are means ± SD. * *p* < 0.05.

**Figure 3 nutrients-13-02910-f003:**
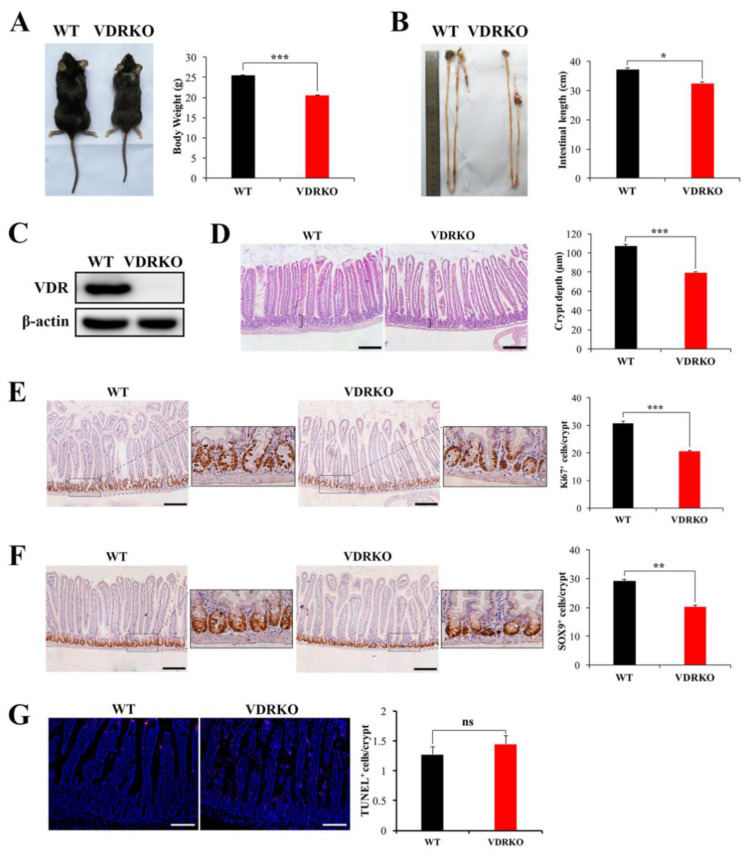
VDR deficiency impairs the intestinal physiological structure and crypt stem/progenitor cell proliferation. (**A**) Appearance of wild type (WT) and VDR knockout (VDRKO) mice and quantification of body weights at the age of 8 weeks. *n* = 12 in each group. (**B**) Appearance of intestines of WT and VDRKO mice and quantification of intestine length. *n* = 7 in each group. (**C**) VDR was evaluated in intestine tissues of mice by WB. (**D**) Representative H&E images of intestine of WT and VDRKO mice and quantification of crypt depth. *n* = 6 in each group. Scale bar, 200 μm. (**E**) Representative IHC images of Ki67 staining of WT and VDRKO mice and quantification of number of Ki67^+^ cells/crypt. Scale bar, 200 μm. *n* = 6 in each group. (**F**) Representative IHC images of SOX9 staining of mice and quantification of number of SOX9^+^ cells/crypt. *n* = 6 in each group. Scale bar, 200 μm. (**G**) Representative IF images of TUNEL staining of WT and VDRKO mice and quantification of TUNEL^+^ cells/crypt. *n* = 6 in each group. Scale bar, 100 μm. Values are means ± SD. * *p* < 0.05, ** *p* < 0.01, *** *p* < 0.001.

**Figure 4 nutrients-13-02910-f004:**
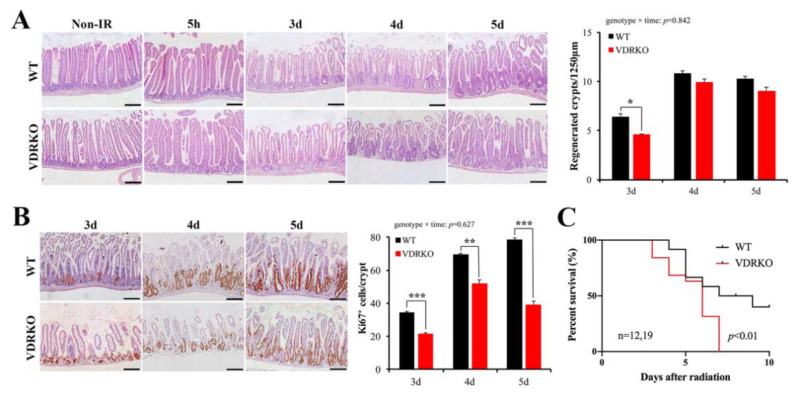
VDR deficiency suppresses intestinal epithelial regeneration and survival time of mice following IR. (**A**) Representative H&E images of intestine of WT and VDRKO mice at different times of 0, 5 h, 3 d (days), 4 d and 5 d after 12 Gy IR. Quantitative analysis of the number of regenerated crypts/1250 μm at different times after IR. *n* = 6 in each group. (**B**) Representative IHC staining of the expression of Ki67 of WT and VDRKO mice and quantification of Ki67^+^ cells/crypt at different times after IR. *n* = 6 in each group. Values are means ± SD. * *p* < 0.05, ** *p* < 0.01, *** *p* < 0.001. Scale bar, 200 μm. (**C**) Kaplan–Meier survival analysis of mice subjected to 12 Gy IR. The statistical analysis was performed by log-rank test. *n* ≥ 12 in each group.

**Figure 5 nutrients-13-02910-f005:**
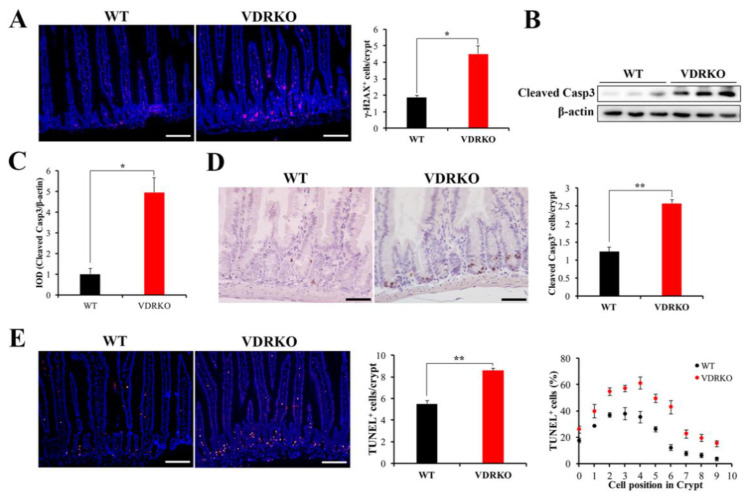
VDR deficiency aggravates IR-induced DNA damage and crypt stem/progenitor cell apoptosis. (**A**) Representative IF images showing the γH2AX expression at 5 h post 12 Gy IR and quantification of γH2AX^+^ cells in crypts. *n* = 6 in each group. Scale bar, 100 μm. (**B**) Cleaved Casp3 was evaluated in intestine tissues of mice at 5 h after IR by WB. (**C**) The Cleaved Casp3 expression was quantitatively analyzed. (**D**) Representative IHC images of Cleaved Casp3 at 5 h after 12 Gy IR and quantification of Cleaved Casp3^+^ cells in crypts. *n* = 6 in each group. Scale bar, 50 μm. (**E**) Representative IF images of TUNEL staining of WT and VDRKO mice at 5 h after 12 Gy IR. Quantification of TUNEL^+^ cells/crypt and their positions in crypts. Scale bar, 100 μm. *n* = 6 in each group. Values are means ± SD. * *p* < 0.05, ** *p* < 0.01.

**Figure 6 nutrients-13-02910-f006:**
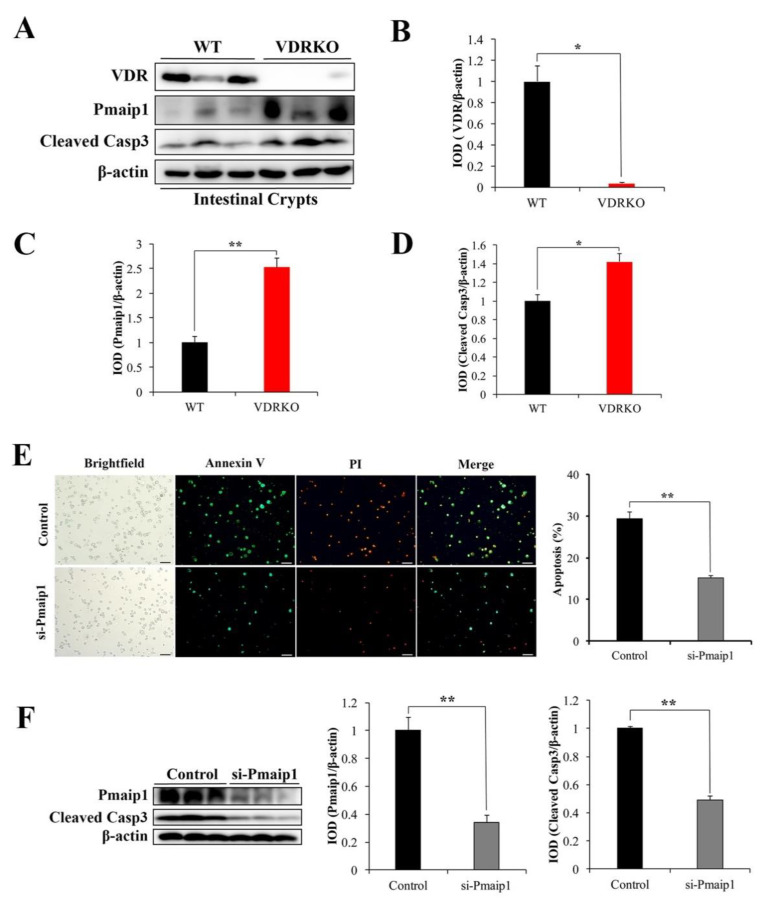
VDR suppresses IR-induced crypt stem/progenitor cell apoptosis via the Pmaip1-mediated pathway. (**A**) VDR, Pmaip1 and Cleaved Casp3 were analyzed in crypt stem/progenitor cells from WT and VDRKO mice at 5 h after IR by WB. (**B**–**D**) Quantitative analysis of WB of VDR, Pmaip1 and Cleaved Casp3 expression, respectively. (**E**) Representative IF images of Annexin V/PI staining at 24 h post 12 Gy IR of IEC-6 cells with or without si-Pmaip1. Quantification of apoptotic cells with Annexin V-positive signaling was analyzed. Scale bar, 20 μm. (**F**) Pmaip1 and Cleaved Casp3 in IEC-6 cells with or without si-Pmaip1 were analyzed at 24 h after IR by WB. Quantitative analysis of WB of Pmaip1 and Cleaved Casp3 expression. Values are means ± SD. * *p* < 0.05, ** *p* < 0.01.

**Table 1 nutrients-13-02910-t001:** The sequences of the primers for RT-qPCR assays.

Gene	Sense (5′-3′)	Antisense (5′-3′)
*Vdr*	GAATGTGCCTCGGATCTGTGG	ATGCGGCAATCTCCATTGAAG
*Gapdh*	AGGTCGGTGTGAACGGATTTG	TGTAGACCATGTAGTTGAGGTCA

## Data Availability

Data presented in this study are available on request from the corresponding author.

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
