# Peer review of "Vitamin D Receptor Protects against Radiation-Induced Intestinal Injury in Mice via Inhibition of Intestinal Crypt Stem/Progenitor Cell Apoptosis"

_nutrients, 2021, doi:10.3390/nu13092910_

Round 1

Reviewer 1 Report

Li et al. present evidence for vitamin D receptor activity in the response of intestinal crypts to ionizing radiation. They use several apoptosis-related markers to implicate cell death in this process. They also offer a preliminary implication of the Pmaip1 protein in the proapoptotic pathway. However, the only functional study of Pmaip1 presented was done if an in vitro IEC-6 cell culture model, which limits the interpretation of their results. Despite the authors claims, the impact of vitamin D as a nutrient was not directly studied here.

Major concerns:

Though present in the mice diet, vitamin D concentration was not modulated in animal feeding experiments. No dose-response was presented. Therefore, it is inappropriate to conclude that vitamin D protects against intestinal injury (title, end of abstract, discussion, conclusion). This reviewer suggests revising the wording of the article to avoid making such broad conclusions that are not supported by the experiments/results presented.

The origin of the single siRNA used against Pmaip1 is not provided. It is not possible to determine the specificity of this tool (isoforms), or if it has off-target effects. Nor is any discussion of these issues provided. The interpretation of the authors that Pmaip1 is a crucial apoptotic mediator in their experiments is therefore called into question.

No functional study of the role of Pmaip1 was done in animals. The “Pmaip1 pathway” was not shown by the authors to influence cell apoptosis in mice exposed to radiation. The conclusions the authors reach in the discussion section should revised to avoid over-interpretation (lines 391-392).

Minor concerns:

Line 31 – Though seeking new nutrients for prevention and mitigation of injury is a worthwhile goal, the logical link hinted at in the text is weak at best. Why focus specifically on nutrients to cure cancer?

Line 34 – sentence fragment, revise wording: [and prevent harmful effects for cancer patients receiving radiation therapy]

Line 43 – Missing verb: [VD (acts) as a ligand…]

Line 56 – Revise phrasing: [source of power]

Animal husbandry: the diet used in this study is not properly referenced. It is not possible to obtain the detailed composition of the diet from the information provided. Were animals fed ad libitum? Also, what light/dark cycle were the animals exposed to?  

The technique for quantification of immunofluorescence, immunohistology or TUNEL results were not explained in the methods section.

The quantification technique of Western blots bands was not explained. What does “IOD” stand for in figures?

Line 209 – “elevated again”; expression is detectable but hardly elevated

Line 214 – “which is a suitable model that emulates VD deficiency”; wording is awkward, and no reference is provided to support the claim. Are all VDR isoforms affected, all VD known effectors?

Line 245 – “As a result”; the logical link is weak

Line 266 – “positions 3-6”; positions 2-4 on the graph are the actual position shown to be more frequent

Line 269 – Missing word: [IR-induced (crypt) stem/progenitor…]

Lines 277 to 279 – The modulation of Fgf1, Fn1, Pdgfb and Igf1 expression is not statistically significant according to the figure provided. It seems inappropriate to say that expression was upregulated/downregulated.

Line 296 – Contrary to what is written, Pmaip1 expression is not presented in figure 6C. The authors might want to mention that lack of VDR expression in their knockout tissue (actual Fig. 6C result).

Line 313 – What statistical test was used for the analysis of survival curves? Are values shown in figure 4C really means and SD?

Line 365 – Upregulation of VDR in the context discussed does not demonstrate its involvement in apoptosis. The logic used by the authors is faulty here. It is their knockout results that show a causal relationship. Maybe it is only the use of the word “demonstrate” that is to be revised.

Reviewer 2 Report

In this report, the authors describe mechanistic studies to explore the role of VD/VDR during epithelium recovery following irradiation injury to the intestine.  Overall, the paper is reasonably well written, and the experiments are sound in design and data interpretation.  I found the work to be of interest, but have a few comments that should be addressed in a revised paper.

The statistical methods need to be revised as appropriate for the experiment design.  First, with three replicated experiments, the biological sample size is just n=3, which does not support use of parametric tests that have assumptions for data normality and equal variance.  Thus, with n=3, the authors must instead use a non-parametric test for their analyses.  For data sets with n = 4 or more, then parametric tests are okay, but the authors need to verify that the data are normally distributed with equal variance; if not, the nonparametric tests are needed.   Second, for some data (e.g., Figure 1b, 1c and Fig 4 ), the experiment design includes two factors (e.g., for Fig 1 irradiation and vitamin D treatment; for Fig 4, time and genotype), and thus requires the appropriate statistical model to account for two factors.  Using a student's t-test is incorrect.  When multiple factors are included, the main effects and interaction should be provided, in addition to any specific post-hoc tests performed.  Also, multiple testing corrections must be applied. The authors are encouraged to consult with a statistician to correct these deficiencies in their data analyses.

For imaging experiments, please provide a bit more detail on the robustness of the data acquisition.  For each experiment, how many fields of view were captured?  How many total cells were assessed?

For figure 1A, the fluorescence for gamma-H2AX is very weak, hard to see in the image. I suggest a global contrast adjustment applied to all images in that column to assist in visualization.  The same can be done - in a global manner - for other images that are difficult for the reader to see.

Data in figure 2C are incorrectly plotted.  The units are given equal distance on the X axis when in fact the difference between 5 hr and 3d is much greater than between 0 h and 5 h.  Please plot these data properly on an XY plot with continuous scales in both axes.

The language used in lines 213-216 suggest that the authors "generated" the VDRKO model, when it seems that these mice were purchased as heterozygotes from a supplier.  This is likely unintentional by the authors, but I suggest a careful consideration here of editing.  Perhaps inclusion of a citation for the original production of the VDRKO mice (if not produced originally by the company) would be appropriate.

The authors attribute the findings presented in figure 4 (reduced cell proliferation as indicated by few Ki67+ cells on days 3, 4 and 5 post IR) to the VDR.   Yet, data in figure 2 suggest that in WT mice, IR triggers a loss of VDR expression, such that the protein was undetectable on days 3 and 4 post IR.   So, how really different are the WT and KO mice during these days in the context of VDR activity?  I'm having a hard time reconciling these two data sets and the statement on line 246-247.

Text in section 3.6 suggests that pathway enrichment analysis was performed following transcriptome analyses.  Where are the full transcriptome data?  Results of the enrichment analyses?

The statement on line 344 that "vitamin D was a protective nutrient" needs refinement in my opinion.  This paper revealed a likely role for vitamin D in intestinal repair following IR.  But, the authors did not test the obvious model with dietary intervention (or other delivery method for active D3) to determine if elevated D3 status would protect against IR damage or speed recovery in WT mice with normal receptor expression.  This missing element to the story limits translation of these findings to real world clinical situations.

The authors failed to discuss any limitations to their work.

The copy editors will probably help, but try to position the figures closer to the text that describes them.

Round 2

Reviewer 2 Report

The authors have been mostly responsive to the reviewers' critiques.

Data in figure 2c absolutely can be plotted as XY scatter plot; just use a consistent unit for the X-axis (hours as 0, 5, 72, 96, 120).  A mock up illustrating how to do this is attached to this review.

The methods for RNAseq are not described in the manuscript, nor methods for data analyses.  Are the p-values presented in Figure S2 FDR-adjusted?  The x-axis labels for plot S1a make no sense at all.  Nor is the scale used for the padj values in figure S1c useful.  Typically, one plots the -log FDR q-value.  Overalll, these RNAseq results are poorly described.  The authors refer to KEGG pathway analysis, but no details are provided as to how that enrichment analysis was performed.  RNAseq data must be deposited in public repository (NCBI Gene Expression Omnibus).  

Looking at the table provided, the PI3K-Akt and p53 pathways do not appear to be enriched at all (value of 0.055 and 0.114 before FDR adjustment!  So, the justification for using these RNAseq data to inform further exploration of these pathways isn't sound.  Indeed the top pathway is associated with Staph infection (raises questions about health of the animals).  And, only three pathways were significantly enriched (none of which were associated with control of transcription, which I find odd for an animal model in which a transcription factor was knocked out), which is indeed a bit surprising and makes me wonder about the processing of the RNAseq data ahead of these enrichment analyses. The authors must provide sufficient detail for methodology and analyses such that another research group could repeat their work.  Likewise, the results need better presentation an interpretation, primarily for the sequencing data.
